# Effect of Aluminum, Iron and Chromium Alloying on the Structure and Mechanical Properties of (Ti-Ni)-(Cu-Zr) Crystalline/Amorphous Composite Materials

**Andrey A. Tsarkov [1,2,]\*, Vladislav Yu. Zadorozhnyy [1,3], Alexey N. Solonin [1] and Dmitri V. Louzguine-Luzgin [2,4]**

[1] National University of Science and Technology (MISIS), Leninsky Prosp, 4, Moscow 119049, Russia; vuz@misis.ru (V.Y.Z.); solonin@misis.ru (A.N.S.)

[2] WPI Advanced Institute for Materials Research, Tohoku University, Katahira 2-1-1, Aoba-Ku, Sendai 980-8577, Japan; dml@wpi-aimr.tohoku.ac.jp

[3] Erich Schmid Institute of Materials Science, Austrian Academy of Sciences, 8700 Leoben, Austria

[4] MathAM-OIL, National Institute of Advanced Industrial Science and Technology (AIST), Sendai 980-8577, Japan

\* Correspondence: andrei.tsarkov@outlook.com; Tel.: +7-495-955-01-34

**Abstract:** High-strength crystalline/amorphous composites materials based on (Ti-Ni)-(Cu-Zr) system were developed. The optimal concentrations of additional alloying elements Al, Fe, and Cr were obtained. Structural investigations were carried out using X-ray diffraction equipment (XRD) and scanning electron microscope (SEM) with an energy-dispersive X-ray module (EDX). It was found that additives of aluminum and chromium up to 5 at% dissolve well into the solid matrix solution of the NiTi phase. At a concentration of 5 at%, the precipitation of the unfavorable $NiTi_2$ phase occurs, which, as a result, leads to a dramatic decrease in ductility. Iron dissolves very well in the solid solution of the matrix phase due to chemical affinity with nickel. The addition of iron does not cause the precipitation of the $NiTi_2$ phase in the concentration range of 0–8 at%, but with an increase in concentration, this leads to a decrease in the mechanical properties of the alloy. The mechanical behavior of alloys was studied in compression test conditions on a universal testing machine. The developed alloys have a good combination of strength and ductility due to their dual-phase structure. It was shown that additional alloying elements lead to a complete suppression of the martensitic transformation in the alloys.

**Keywords:** composite materials; mechanical properties improvement; structure–property relations; metallic glass composite; amorphous materials; martensite phase

## 1. Introduction

Numerous studies show that bulk metal glassy (BMG) alloys have significantly higher mechanical properties than their crystalline analogs [1,2]. However, due to the peculiarities of their atomic structure, this class of materials is prone to brittle fracture and has extremely low ductility. Creating a composite material based on crystalline and amorphous phases could help to avoid the disadvantages of its constituent components [3–8].

One way to significantly increase the level of ductility is to use the phenomenon of the Transformation-Induced Plasticity (TRIP) effect [9–14]. This effect is well studied and observed in various systems [15–21]. During deformation, the austenitic phase turns into the martensitic phase.

This transformation contributes to the redistribution of stresses inside the sample, which is the reason the unique mechanical behavior of these alloys [22–28]. The TRIP effect has been discovered on alloys of the NiTi system and, at the moment, this binary system is the most studied [29,30].

The aim of our study was to develop a new composite material, within which will combine the high strength of metal glass and the good ductility of a crystalline alloy [31–34]. We continued to work with the system (Ti-Ni)-(Cu-Zr), which showed good potential [4,21]. In this work, we studied the effect of alloying by Al, Fe and Cr on the structure and mechanical properties of our alloy. These additives were selected because of their positive effect on the structure [35–38].

The compositions of the alloys were chosen based on the consideration that the alloying element tends to be a substitute for electronically similar atoms. Zr has similar electron configuration in the outer shells to Ti. Therefore, it is expected that Zr will be a substitute for Ti, rather than Ni. Fe and Co are close to Ni in the periodic table; thus, it is expected that Fe and Co will be substitutes for Ni. Nakata et al. conducted several studies to determine the site occupancy of the alloying element in Ti-Ni alloys [39,40]. They found that Fe and Co prefer to be substitutes for Ni, while Cr and Cu seem to prefer to be substitutes for both Ti and Ni. Based on these results, we chose several compositions in which Fe and Cr were added replacements for Ni, and Al was added by reducing the total content of nickel and titanium, because Al tends to be a substitute for Ni and Ti.

## 2. Materials and Methods

For alloy preparation, pure (not less than 99.99%) metals were used. The compositions of the studied alloys are presented in Table 1. Master alloys were smelted in an arc-melting machine (custom made) under an argon atmosphere with a Ti getter. After all raw materials were melted together, the obtained ingot was flipped and remelted. After four remelts, the ingot acquired a homogeneous chemical composition. Samples of composite materials were obtained by rapid cooling into a massive copper mold by the ejection casting technique. The obtained samples are 3 mm in diameter and 50-mm long. The base system was chosen from previous works [6,27,41,42]. The concentrations of alloying elements were selected to preserve the equiatomic composition of the nickel and titanium in the alloy.

**Table 1.** Chemical composition of the studied alloys (in atomic percentage).

| Alloy | Ti | Ni | Cu | Zr | Co | Y | Al/Fe/Cr |
|-------|-------|-------|----|----|----|-----|----------|
| Al-1  | 40.25 | 40.25 | 8  | 8  | 2  | 0.5 | Al 1     |
| Al-2  | 39.75 | 39.75 | 8  | 8  | 2  | 0.5 | Al 2     |
| Al-3  | 39.25 | 39.25 | 8  | 8  | 2  | 0.5 | Al 3     |
| Al-5  | 38.25 | 38.25 | 8  | 8  | 2  | 0.5 | Al 5     |
| Cr-1  | 40.75 | 39.75 | 8  | 8  | 2  | 0.5 | Cr 1     |
| Cr-2  | 40.75 | 38.75 | 8  | 8  | 2  | 0.5 | Cr 2     |
| Cr-3  | 40.75 | 37.75 | 8  | 8  | 2  | 0.5 | Cr 3     |
| Cr-5  | 40.75 | 35.75 | 8  | 8  | 2  | 0.5 | Cr 5     |
| Fe-1  | 40.75 | 39.75 | 8  | 8  | 2  | 0.5 | Fe 1     |
| Fe-2  | 40.75 | 38.75 | 8  | 8  | 2  | 0.5 | Fe 2     |
| Fe-3  | 40.75 | 37.75 | 8  | 8  | 2  | 0.5 | Fe 3     |
| Fe-5  | 40.75 | 35.75 | 8  | 8  | 2  | 0.5 | Fe 5     |
| Fe-8  | 40.75 | 32.75 | 8  | 8  | 2  | 0.5 | Fe 8     |

The structure of the specimens was studied by the X-ray diffraction technique and scanning electron microscopy. XRD analysis was carried out with monochromatic Cu-K$\alpha$ radiation using a Bruker D8 diffractometer (Billerica, MA, USA). The dimensions of the samples for structure investigation were 3 mm in diameter and 1-mm thick. XRD tests were carried out in a range of diffraction angles, 2$\theta$ 30–80°, with a step of 0.02° and an exposure time of 10 s. Total peak, which is a sum of amorphous and crystalline peaks, was separated into two initial peaks by using OriginLab software. The volume fraction of the amorphous phase was estimated by the intensity ratio from the total and pure amorphous peaks.

SEM analysis was performed using TESCAN VEGA LMH (Brno, Czech Republic) in back-scattered electrons (BSE) mode with cathode $LaB_6$. For EDX analysis, we used the X-Max 80 detector (Oxford Instruments, Abingdon, United Kingdom).

Mechanical tests of samples were carried out under compressive conditions using an Instron 5581 universal testing machine (Norwood, MA, USA). The dimensions of the samples for mechanical tests were 3 mm in diameter and 6-mm long. The deformation rate was $5 \times 10^{-4}$ s$^{-1}$.

## 3. Results

### 3.1. Structure

The results of the microstructure investigation of the as-cast samples using the XRD method are shown in Figure 1. According to the obtained X-ray diffraction patterns, it can be concluded that the structure of the samples mainly consists of the NiTi phase. The sample mostly consists of the crystalline phase of NiTi cP2. In addition, an amorphous phase is also present in the structure, the presence of which is confirmed by a diffuse broad peak in the range 2θ 37–46°. For Al and Cr, the maximum allowable concentration is 5 at%, and for Fe it is 8 at%. With an increase in the content of the additional alloying element, the tendency to precipitate the $NiTi_2$ phase is well documented. This is due to the chemical affinity of Fe and Ni. Because of this, the Fe atoms can easily occupy the Ni atoms' positions and not cause the precipitation of intermetallic phases.

Figure 2 shows the structure of the as-cast samples, obtained using the SEM technique. The structure mainly consists of two phases, which form a mesh-like structure. The largest fraction is the crystalline NiTi phase (dark areas), along the borders of which there is an amorphous phase (grey-white areas). It can be noted that a small third phase exists, which is presented in the form of very bright dots inside the amorphous phase. We were not able to identify this phase because of its small size. The average phase size was about 1 μm, which is not enough to determine the chemical composition by the EDX method. This phase was not detected by XRD in any of the samples, which indicates its exceedingly small volume fraction. Such a low volume fraction does not affect the mechanical behavior of the sample.

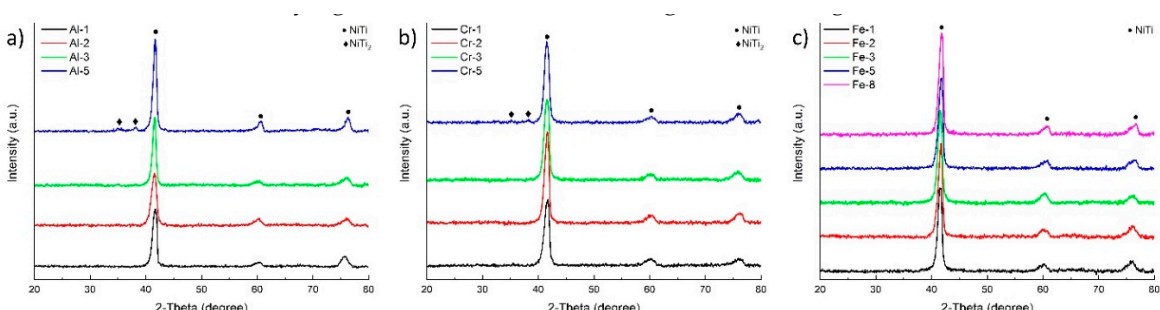

**Figure 1.** XRD patterns of as-cast samples: (**a**) alloys doped with Al, (**b**) alloys doped with Cr, (**c**) alloys doped with Fe.

It should be noted that there was a rather large volume of pores, located mainly along the borders, caused by the casting method. However, as the volume fraction of the pores was not large (much lower than 1%), there was no effect on the mechanical properties. The volume fraction of the amorphous phase in the samples was about 20–25%. All alloys have a similar morphology of structure. Additional alloying elements did not lead to significant changes in the microstructure.

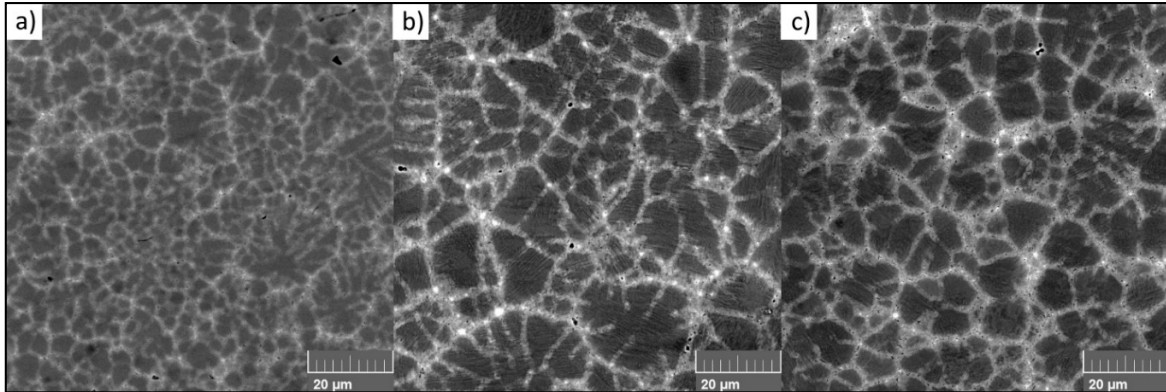

**Figure 2.** SEM images of as-cast samples (BSE mode): (**a**) alloy Al-1, (**b**) alloy Cr-1, (**c**) alloy Fe-1.

To determine the chemical composition of the phases, the EDX method was used. The average chemical composition of the phases is shown in Figure 3. According to the obtained data, it can be concluded that the crystalline phase is a highly alloyed solid solution of the cP2 NiTi phase. The composition of the amorphous phase almost completely coincides with the average composition of the alloy. The only exception is Co, the content of which in the amorphous phase is much lower than in the crystalline one. The average chemical composition measured by the EDX method is close to the corresponding alloy composition. The concentration of the studied areas is given in Table 2. EDX analysis shows an increase in the aluminum content in both amorphous and crystalline phases, with an increase in the aluminum content in the alloy composition, while the aluminum content in the amorphous phase is higher than that in the crystalline one. This is because aluminum atoms are much easier to fit into the amorphous phase than into the ordered crystalline one.

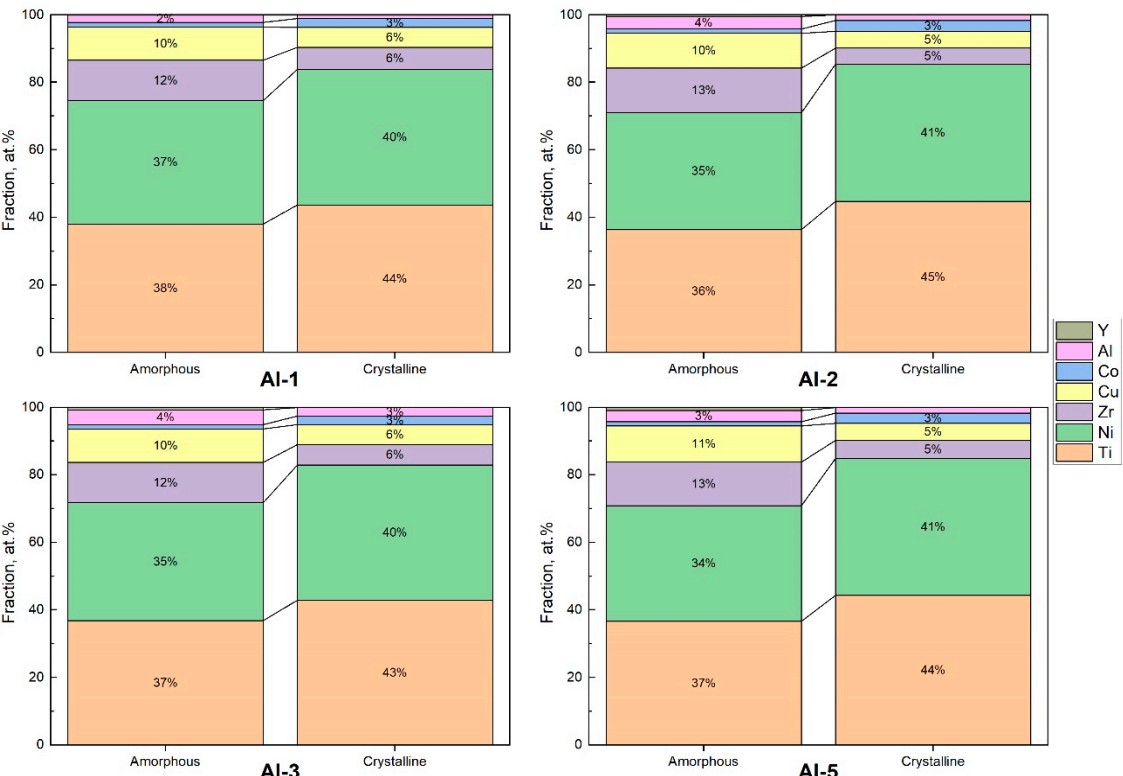

**Figure 3.** Chemical composition of amorphous and crystalline phases studied by EDX for alloys with Al addition (in atomic percentages).

**Table 2.** Chemical composition of amorphous and crystalline phases studied by EDX for alloys with Al addition (in atomic percentages).

| Alloy | Phase | Ti | Ni | Cu | Zr | Co | Y | Al |
|-------|-------|------|------|------|------|-----|-----|-----|
| Al-1 |           | 38.1 | 36.5 | 9.9  | 11.9 | 1.4 | 0.1 | 2.2 |
| Al-2 | Amorphous | 36.4 | 34.7 | 10.3 | 13.2 | 1.2 | 0.5 | 3.7 |
| Al-3 |           | 36.8 | 34.9 | 9.9  | 12.0 | 1.3 | 0.7 | 4.4 |
| Al-5 |           | 36.6 | 34.2 | 10.7 | 13.0 | 1.2 | 0.9 | 3.3 |
| Al-1 |           | 43.7 | 40.3 | 5.9 | 6.5 | 2.7 | 0.1 | 0.9 |
| Al-2 | cP2       | 44.7 | 40.7 | 4.9 | 4.8 | 3.3 | 0.0 | 1.6 |
| Al-3 |           | 42.9 | 40.1 | 5.9 | 6.0 | 2.6 | 0.0 | 2.6 |
| Al-5 |           | 44.3 | 40.5 | 5.0 | 5.4 | 3.1 | 0.0 | 1.7 |

### 3.2. Mechanical Behavior

This work is a continuation of the study in [27]. In the previous work, most of the studied alloys underwent a martensitic transformation. Due to martensitic transformation, alloys exhibit a high ductility (for most alloys, this is more than 20%). The strength of the alloys in both works is remarkably high (more than 2000 MPa). However, the martensitic transformation also has a negative effect in terms of potential applications. Deformation in martensitic alloys begins at a low level of stress, which induces martensitic transformation ($\sigma_M$). From the point of view of practical applications, this is an undeniable disadvantage. It is important to have a monotonous character of deformation. The alloys obtained in this work are a result of the optimization of compositions of previously studied alloys. The resulting alloys have more attractive characteristics for practical use.

In most of the studied alloys, the additional alloying elements led to the suppression of martensitic transformation, which was observed in [10,27,43]. The only exception is alloy Al-1. The deformation curves for alloys with and without martensitic transformation are shown in Figure 4. The compression curve of alloy Al-1 consists of three stages. The first stage is elastic deformation, the second is superelasticity caused by martensitic transformation, and the third is plastic deformation. For other alloys, the deformation curve consists of two stages: elastic and plastic deformation.

According to compression curves (Figures 5–7) it can be concluded that an increase in the content of alloying elements in most cases leads to a decrease in ductility and ultimate strength, and, at the same time, increases yield strength. This is a consequence of the increasing distortion of the atomic structure. The values of ultimate stress ($\sigma_b$), yield stress ($\sigma_{0.2}$), stress of onset plastic deformation ($\sigma_R$), stress that induces martensitic transformation ($\sigma_M$) and plasticity ($\varepsilon$) are shown in Table 3.

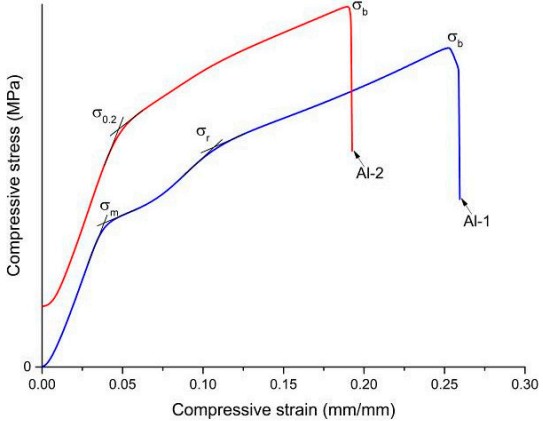

**Figure 4.** Stress–strain curves (schematic) for alloys with and without martensitic transformation (Al-1 and Al-2 correspondently).

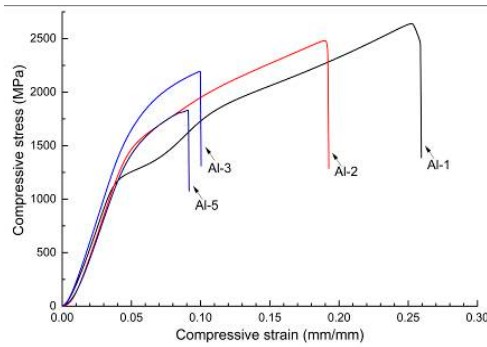

**Figure 5.** Stress–strain curves of alloys doped with Al.

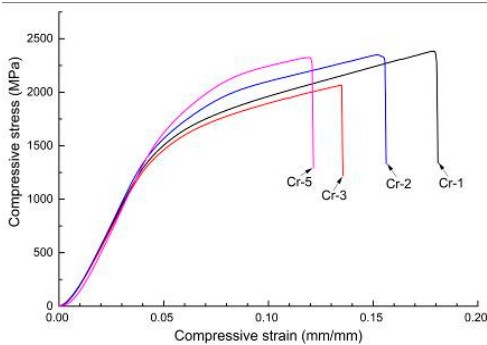

**Figure 6.** Stress–strain curves of alloys doped with Cr.

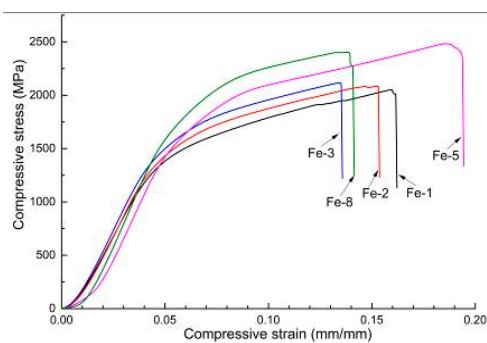

**Figure 7.** Stress–strain curves of alloys doped with Fe.

It is necessary to mention the possible presence of the R phase in the structure. The R phase has been extensively studied. Unlike the mP4 phase (herein after M), the R phase has a rhombohedral lattice [44,45]. The R phase is less stable than M. At the same time, the energy barrier to its formation is lower than that of the M one. During cooling, the R phase precipitates first (if there are no other external factors) [46–48]. However, under the applied load, in most cases, the A→R transformation is suppressed, and the A→M transformation occurs [49,50]. In the present study, the presence of the R phase in the structure is unlikely, or its content is so small that may not even be possible to determine its existence through XRD. The strongest peaks of the R phase are at the angles of ~45 and 52 two theta (in addition to coincident peaks from the matrix NiTi phase). In this work, such peaks are not observed in any of the studied alloys, which indicates the absence of the R phase in the structure of all alloys.

The partial suppression of martensitic transformation is caused by the local distortion of the atomic lattice of the R phase. This distortion leads to the formation of a metastable intermediate phase between cP2 and R. This assumption was made by J.G. Niu and W.T. Geng in [51]. Another group of authors concluded that Fe addition stabilizes the austenitic phase [52]. It seems that the embedded chromium and aluminum atoms have a similar effect on the cP2 phase. The evidence in favor of this idea is a systematic decrease in plasticity with an increasing concentration of alloying elements.

**Table 3.** Mechanical properties of the studied alloys.

| Alloy | $\sigma_{0.2}$, MPa | $\sigma_b$, MPa | $\varepsilon$, % |
|---|---|---|---|
| Al-1 | here $\sigma_M$ 1175; $\sigma_R$ 1750 | 2650 | 25.2 |
| Al-2 | 1575 | 2490 | 18.9 |
| Al-3 | 1790 | 2190 | 10.1 |
| Al-5 | 1670 | 1830 | 9.1 |
| Cr-1 | 1590 | 2390 | 17.9 |
| Cr-2 | 1635 | 2360 | 15.2 |
| Cr-3 | 1520 | 2070 | 13.5 |
| Cr-5 | 1970 | 2330 | 11.9 |
| Fe-1 | 1390 | 2060 | 15.9 |
| Fe-2 | 1550 | 2110 | 15.2 |
| Fe-3 | 1580 | 2120 | 13.5 |
| Fe-5 | 1810 | 2510 | 18.7 |
| Fe-8 | 2030 | 2410 | 13.9 |

## 4. Discussion

An increase in the content of additional alloying elements leads to a systematic increase in the yield strength and a decrease in ductility. Such a change in mechanical behavior is associated with lattice distortion near the atoms of the additional alloying elements. The distortion impedes the movement of dislocations in the cP2 crystalline phase [53,54]. The addition of iron does not change the mechanical behavior of the alloy significantly. With an increase in the iron content, a significant decrease in ductility does not occur. As noted earlier, iron atoms are very similar to nickel atoms in size, which allows them to easily occupy their positions in the NiTi lattice [55,56]. Since the number of iron atoms is much smaller in the alloy, such a replacement does not lead to a strong distortion of the atomic lattice. The addition of chromium leads to expected changes in the mechanical properties: an increase in strength and a decrease in ductility.

A significant drop in the ductility of the Al-5 alloy is caused by the presence of the $NiTi_2$ phase [57–59]. Alloy Al-3 also demonstrated a dramatic decrease in mechanical properties, but the presence of the NiTi2 phase was not detected in the X-ray diffraction pattern. It can be concluded that the $NiTi_2$ phase is also present in alloy Al-3, but its amount was not detected by the X-ray method. Nevertheless, even a small NiTi2 phase leads to a strong decrease in the mechanical properties of the alloy.

Additional studies were carried out to confirm the presence of martensitic transformation in the Al-1 alloy. Al-1 and Al-2 alloys were loaded to a stress level of 2000 MPa, then the load was removed, and the alloys were investigated using XRD analysis. A martensitic transformation with similar compression curves was observed in alloys of a similar system in [4,9,60,61]. Figure 8 shows an XRD pattern for Al-1 alloy. There is a significant change in the XRD pattern. The main peak (at an angle of 42 degrees) has a distinct broadening. This is caused by the formation of a peak shoulder due to NiTi's martensitic phase (mP4 phase M). Similar changes in the structure were observed in earlier works [9,10,12,43,62–64]. Figure 9 shows an XRD pattern for the Al-2 alloy. Its pattern is almost identical to that of the deformed state. A slight shift in the diffraction peaks is observed in the deformed state. This, combined with the mechanical test results, is evidence of the presence of partially suppressed martensitic transformation in the Al-1 alloy. Moreover, for all other alloys, martensitic transformation is fully suppressed.

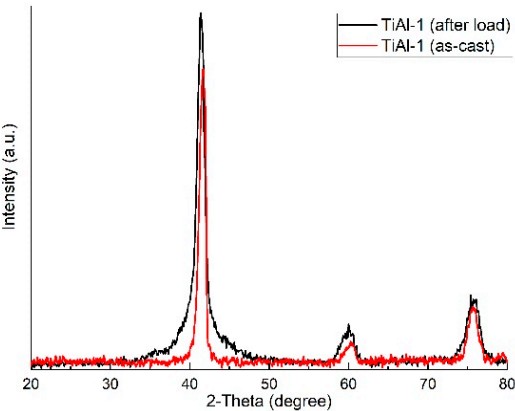

**Figure 8.** XRD pattern of alloy Al-1 in as-cast and after loading.

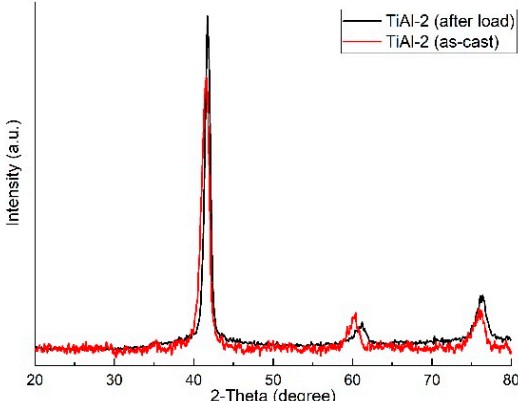

**Figure 9.** XRD pattern of alloy Al-2 in as-cast and after loading.

## 5. Conclusions

The additions of Al, Fe and Cr to the NiTiCuZr glassy/cP2 crystal alloy studied earlier lead to an increase in the yield strength. The ultimate strength and plasticity of most samples exceed 2000 MPa and 15%, respectively, due to the stabilization of austenite and the suppression of the transformation-induced martensitic transformation.

The addition of more than 5 at% of Al and Cr leads to the precipitation of the $NiTi_2$ phase and a reduction in plasticity.

The addition of Fe does not cause significant structural changes up to 8 at% due to the small difference in atomic sizes and its large solubility in Ni.

**Author Contributions:** A.A.T. performed most of the experiments (sample preparation, mechanical tests, XRD) and wrote the manuscript, V.Y.Z. performed the microstructure investigations (SEM) and contributed to the interpretation of the results, A.N.S. planned the experiments and provided equipment, D.V.L.-L. conceived the original idea and led the research. All authors discussed the methods, results, and participated in the writing and checking of the manuscript. All authors have read and agreed to the published version of the manuscript.

**Funding:** This work was supported by the World Premier International Research Center Initiative (WPI), Ministry of Education, Culture, Sports, Science and Technology (MEXT), Japan, by the Ministry of Education Science of the Russian Federation in the framework of the Increase Competitiveness Program of National University of Science and Technology MISiS (№ K2-2020-006 and K2-2014-013). The authors gratefully acknowledge the Austrian Science Fund (FWF) under project grant I3937-N36 and the international research collaboration of ICC-IMR (Tohoku University, Japan) and ESI ÖAW (Austrian Academy of Science), in the framework of the Integrated Project "International joint research on development of material science", grant number 2019PJT2.

**Acknowledgments:** Andrey A. Tsarkov sincerely thanks Andrey A. Stepashkin for his kind technical support with the mechanical test experiments.

**Conflicts of Interest:** The authors declare no conflict of interest. The funders had no role in the design of the study; in the collection, analyses, or interpretation of data; in the writing of the manuscript, or in the decision to publish the results.

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
