# Peer review of "Effect of Aluminum, Iron and Chromium Alloying on the Structure and Mechanical Properties of (Ti-Ni)-(Cu-Zr) Crystalline/Amorphous Composite Materials"

_metals, doi:10.3390/met10070874_

Round 1

Reviewer 1 Report

The investigation reported in this manuscript is a continuation of a former research already published by some of the authors in 2016, (ref. 27).

  • The interest of this kind of study can be beyond the average, but some details should be better described in order to have a more complete understanding of the metallurgical causes behind the reported results:
  • The % of Ti is kept constant in the Cr- and Fe- based combinations, but not in the Al- type alloys. Which is the reason for this choice?
  • Three different types of compositions have been considered, but it seems that only the Al-based composition has been fully studied by XRD and EDS. Are there differences in the case of the Cr- and Fe- based compositions?
  • How have the authors computed the % of amorphous phase and which is the accuracy of the compositional analysis of each amorphous or crystalline phase?
  • In the conclusions it is said that “Addition of more than 5 at.% of Al and Cr leads to precipitation of NiTi2 phase and reduction of plasticity”, but no compositions beyond this 5% have been tested in this paper.

In addition, the text is, perhaps, too schematic. Sections 1 and 2 would admit a more detailed development.

Reviewer 2 Report

The paper is technically interesting and present an interest to the journal readers. Engineering part in the paper is in relatively good order. Analysis and scientific justification are much weaker.

The abstract needs to be re-written and to be factual instead of proclamations

There are some slips in language. For example I do not know what "Vladislav Yu. Zadorozhnyy performed state-of-the art and some experiments" means.

SEM analysis needs explanation. I guess that imaging was done in BSE mode, which needs to be mentioned in Fig.2 caption. Also, there are much more phases as just two authors are trying to analyse. I appreciate that the system is very complex but the authors should openly admit it
